# Is It Time to Supersede the Diagnostic Term "Melanoma In Situ with Regression?" A Narrative Review

Anna Colagrande, Giuseppe Ingravallo * and Gerardo Cazzato *

Section of Molecular Pathology, Department of Precision and Regenerative Medicine and Ionian
Area (DiMePRe-J), University of Bari "Aldo Moro", 70124 Bari, Italy
* Correspondence: giuseppe.ingravallo@uniba.it (G.I.); gerardo.cazzato@uniba.it (G.C.);
  Tel.: +39-3405203641 (G.C.)

**Abstract:** Traditionally, the term melanoma in situ (MIS) is used to designate a horizontal (radial) growth phase of malignant melanoma (MM) in which there is no histological evidence of any invasion (or microinvasion) of neoplastic melanocytic cells into the superficial or papillary dermis. In daily dermatopathological practice, we are faced with misleading definitions, such as "melanoma in situ with regression," which risk affecting homogeneity for comparison purposes of pathological reports of malignant melanoma. The authors conducted a literature review using PubMed and Web of Science (WoS) as the main databases and using the following keywords: "Malignant Melanoma in situ" or "Melanoma in situ" and "regression" and/or "radial growth phase regression." A total of 213 articles from both analyzed databases were retrieved; finally, only eight articles in English were considered suitable for the chosen inclusion criteria. In consideration of the absence of studies with large case series, of reviews with meta-analyses, and, therefore, of a broad scientific consensus, expressions including "melanoma in situ with regression" should be avoided in the histopathological report. Instead, they should be replaced with clearer and more exhaustive definitions.

**Keywords:** melanoma in situ; MIS; differential diagnosis; dermatopathology; malignant melanoma; regression; radial growth phase regression





## 1. Introduction

Malignant melanoma (MM) constitutes the "great mime" of human pathology, with characteristic and different variants that make its recognition particularly difficult [1]. Furthermore, as incidence and prevalence rates increase, it becomes particularly important to identify all the parameters useful in staging the MM patient, as this is fundamental to planning the correct therapy [2]. It is essential that histological reports note elements including Breslow thickness [3] (Clark's level, however, is no longer to be reported), the presence of tumour lymphocytic infiltrate (TILs) according to the College of American Pathologists' (CAP) guidelines [4], ulceration [5], the number of mitoses/mmq of neoplasm [6], possible lymphovascular invasion [7] and neurotropism [8], presence of regression [9], and microscopic distance from the lateral margins of surgical resection [10]. Traditionally, the growth of MM is considered in horizontal (radial) and vertical growth phases, depending on the natural history of the neoplastic lesion [11]. Regression, first described and defined in 1953 [12], constitutes one of the most important, but at the same time most debated, parameters among the MM parameters [13]. The College of American Pathologists (CAP) defines histological regression as replacement of tumour cells by lymphocytic inflammation with attenuation of the epidermis and nonlaminated dermal fibrosis with inflammatory cells, melanophagocytosis and telangiectasia [4], but the criteria by which regression has been defined over time have been far from homogeneous or concordant [13]. For example, historically, one of the simplest systems for defining regression was related to present vs. absent, and if present, it could be marked as partial or complete, considering that

according to Shai et al. only <0.27% of primary cutaneous melanomas undergo complete regression [14]. With the passage of time and with different authors proposing different systems for defining and measuring regression, the current CAP guidelines suggest reporting the presence/absence of regression and, in the case of presence, using the 75% cut-off [4]; this modality allows, to some extent, for limiting the inter-observer variability that is historically very high in relation to this parameter. In this regard, it is equally important to consider that the regression parameter can be difficult to evaluate due to strictly technical issues of histological preparations [15]; for example, skin sections that are not perfectly aligned or preparations performed with uneven thicknesses, together with potential technical artefacts, result in a greater difficulty in defining and homogenising regression data, complicating even more a field where the difficulty of agreement is rather high. Various authors have endeavoured to understand the possible etiopathogenesis mechanisms underlying regression. In fact, it has been understood that the major players responsible for regression events are CD8+ T lymphocytes [15], together with CD4+ T lymphocytes and cytokines released by T-helper 1 (Th1) lymphocytes that stimulate, in the tumour microenvironment (TME), the attack and destruction of melanocytic tumour cells [16,17]. A recent study on the important molecule known as CEACAM1 [18], which is linked to increased melanoma cell invasion and migration, found that thin melanomas with higher levels of CEACAM1 overexpression were more invasive. The researchers discovered that tumour cells exhibit less CEACAM1 expression in regions of regression, which may be related to the presence of natural killer cells (NK).

To better understand the mechanism of action, researchers have also looked at the expression of tissue inhibitors of metalloproteinases (TIMP) in melanomas that have regressed. In contrast to those of melanomas demonstrating segmental regression, TIMP1 has been observed to be overexpressed in nonregressing components of melanomas with partial regression. Compared to melanomas demonstrating segmental regression and melanomas without regression, melanomas with partial regression have nonregressing components that are overexpressed with TIMP2. TIMP2 is thought to block the Wnt/catenin pathway, which is supposed to inhibit the proliferation of melanoma cells, although the function of TIMP1 on carcinogenesis is less clear. It is possible that there are immunohistochemical distinctions between partial, segmental, and total regression based on the varying TIMP expression profiles in melanomas of various regression subtypes [19].

The term "melanoma in situ" (MIS) is used to designate a horizontal (radial) growth phase of malignant melanoma (MM) in which there is no histological evidence of any invasion (or microinvasion) of neoplastic melanocytic cells into the superficial or papillary dermis [20]. In recent years, the literature has indicated a greater incidence of MIS diagnoses than in the past, raising the age-old question of whether this datum may be due to an effective increase in the number of incident cases or to a refinement of diagnostic techniques that are now even more powerful [21,22]. On the other hand, regression in MM continues to arouse strong debate within the scientific community, where different studies have reported discordant data [13].

The recent guidelines released by the Italian Society of Pathological Anatomy and Cytology (SIAPEC)-IAP [23] analyzed in detail the studies present in the literature, mainly highlighting the contrasting results, with some papers [24–27] reporting a reduced survival rate (particularly described in thin melanomas), while other papers, such as those by Morris et al. [28] and Ribero et al. [29], demonstrated better survival in cases of MM with regression phenomena.

However, the impact of regression in the MIS context has not yet been extensively studied, and this paucity of knowledge is reflected in routine dermatopathology practice, where different ways of writing the pathology report at different institutions and even within the same working reality reduce the homogeneity of the diagnoses. This has an important impact on the rest of the diagnostic, therapeutic, and assistance pathway for patients affected by MM, and, last but not least, on the outcomes.

In this review, we briefly summarize the current knowledge about the relationship between MIS and regression and consider useful prospects on the basis of the results obtained so far.

## 2. Materials and Methods

The authors conducted a literature review using PubMed and Web of Science (WoS) as the main databases and using the following keywords: "Malignant Melanoma in situ" or "Melanoma in situ" and "regression" and/or "radial growth phase regression." During the review process, various types of scientific articles (original articles, case reports, case series, reviews, communications, and editorials) were included in the study, relating only and exclusively to the relationship between MIS and regression. Studies dealing with histopathological features were analyzed and included. Conversely, studies that did not specifically address MIS with the regression parameter were excluded, and thus the number of included articles was reduced. Finally, we checked books that might cover the topic according to the adopted inclusion criteria.

English-language articles published up to 20 November 2022 were included. In addition, articles and reviews that were mentioned in these publications were used to supplement information on histologic regression.

A summary of the article selection process is reported in Figure 1.

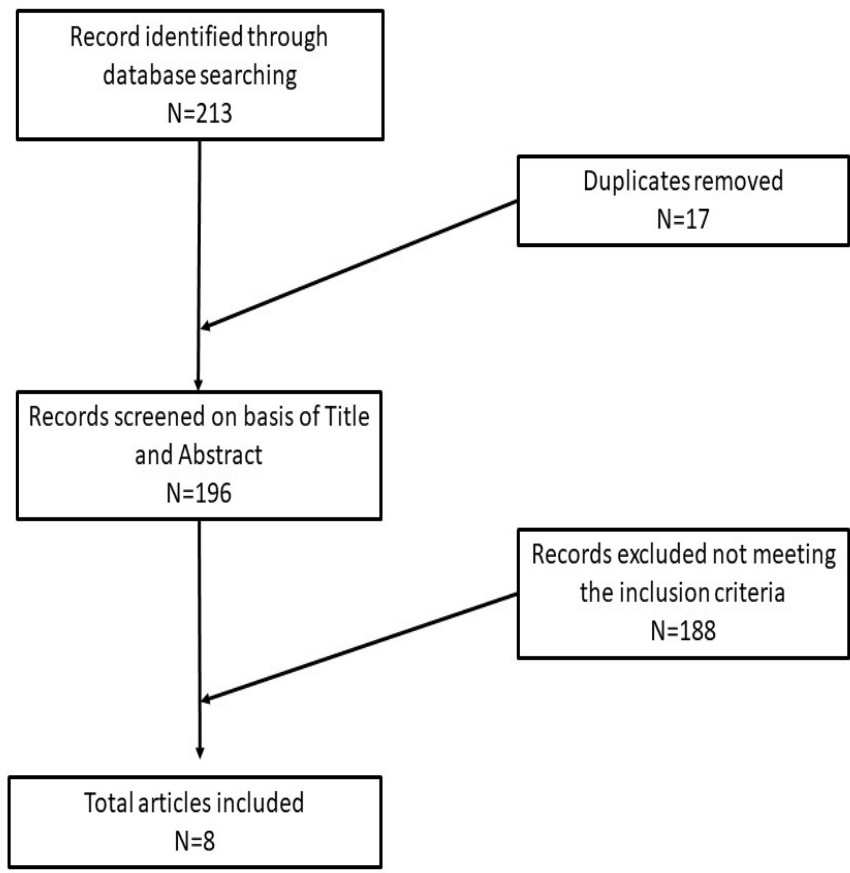

**Figure 1.** Article selection process of this review.

## 3. Results

A total of 213 articles from both analyzed databases were retrieved; finally, only eight articles in English met the chosen inclusion criteria [26–28,30–34].

### 3.1. Histological Features of MIS and Regression

According to the American Joint Committee on Cancer 8th edition (AJCC), MIS is defined as a melanoma that is limited to the epidermis [30] and epithelial adnexal structures, without any microinvasion of the papillary dermis. The latest volume of the "AFIP Atlas of Tumor Pathology, Melanocytic Tumors of the Skin", also defines MIS as a melanocytic lesion confined to the epidermis constituting one of the two types of "radial growth phase melanoma" and distinguishes it from microinvasive melanoma [31]. The difficulty in formulating objective diagnostic criteria for lesions, such as MIS, has been extensively addressed in the literature, as stated in the book by Barnhill [35], in which the author clearly declares that the absence of equal architectural criteria for all means that "some in situ melanomas are dysplastic nevi for other pathologists, and some dysplastic nevi become in situ melanoma for others." In the same book, the importance of adequate sampling is underlined for lesions that may appear to be MIS but that could present micro-invasive areas in further parts of the sample. In any case, in good practice, the criteria most frequently cited in the diagnosis of MIS are: (1) size, even if some melanomas may have a maximum diameter of 2–3 mm; (2) asymmetry of the MIS, which, commonly but not always, is poorly circumscribed compared to a putative junctional nevus; (3) architectural complexity of the MIS lesion compared to a benign lesion, with a greater melanocyte cell plemorphism; and (4) pagetoid spreading, as an MIS will present a more constant and uniform spread of melanocytes than some forms of severe dysplastic nevus do [35].

It is also important to remember that the MIS presents different morphological aspects, as described in detail in the book by Massi and Leboit [32]. The underlying gene alterations, already present in the intraepidermal phase of malignant melanoma, could be the basis of the different histopathological features found in MIS [32]. In fact, MIS may present particular histological features, consisting, in the case of lentigo maligna melanoma (LMM), of atypical melanocytes at the dermoepidermal junction, which, in the initial stages of the lesion, present only a minimal atypia and whose correct recognition constitutes what Plaza et al. define (in their book "Pathology of Pigmented Skin Lesions") as "one of the most difficult challenges for a dermatopathologist" [33] due to the phenomenon of typical/atypical melanocytic hyperplasia both in skin exposed to UV radiation and on previous scarring areas. As LMM progresses, individual melanocytes coalesce along the junction and may extend to the hair adnexa, showing pagetoid spreading and true nests [33]; at this stage, the diagnosis is rather simple [33]. On the other hand, a lesion characterized by the presence of numerous nests of rather uniform melanocytes inside, but with a significant morphological discrepancy of the nests and a different, irregular distance between one and the other, will be more compatible with a nested-pattern MIS [32,33].

Furthermore, the histopathological evaluation of MIS can be very challenging, even for highly experienced pathologists [36], and, at times, the use of ancillary techniques of immunohistochemistry (IHC) can be very useful. Markers, such as Melan-A, HMB-45, SOX-10, and PRAME, assist the dermopathologist on a daily basis in the differential diagnosis between severe dysplastic nevus and MIS. Two examples of MIS are reported in Figures 2 and 3.

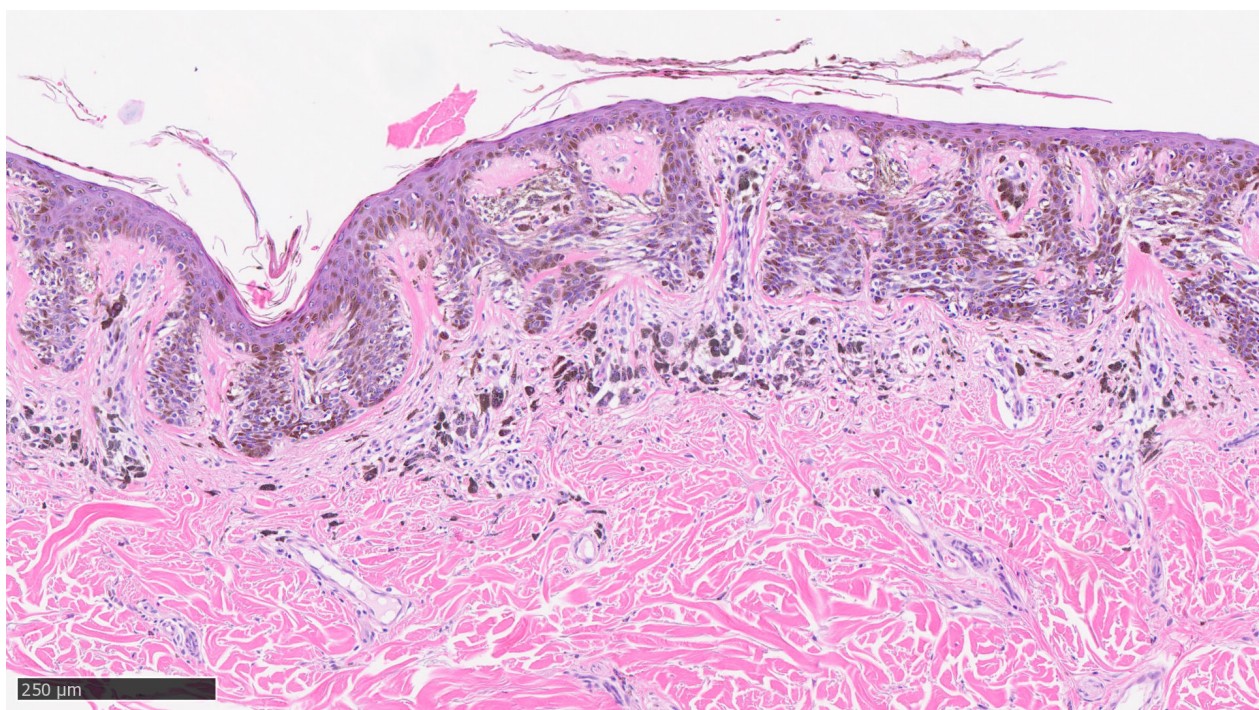

**Figure 2.** Example of melanoma in situ with regression: note the presence of neoplastic melanocytes at the dermoepidermal junction, with presence of sub-epithelial features indicative of regression, such as fibrosis, lymphocytic infiltrate, some clustering melanophages, and neoangiogenesis phenomena (hematoxylin–eosin, original magnification 4×).

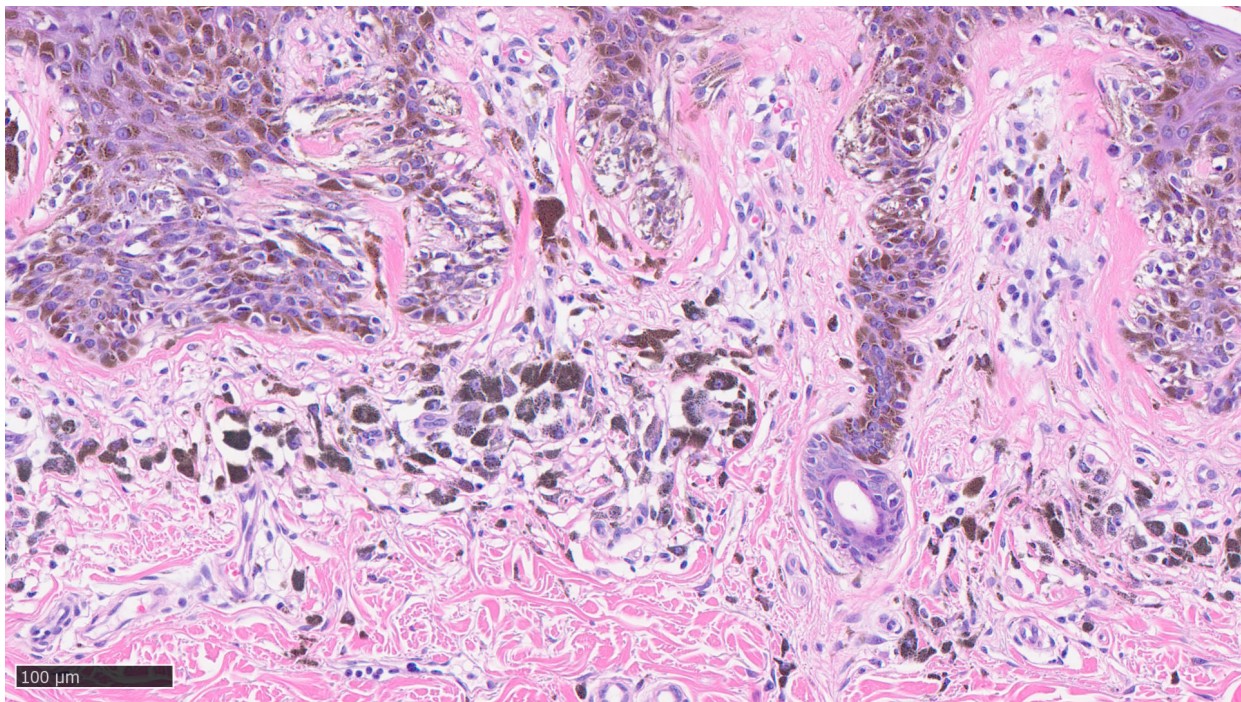

**Figure 3.** Details of previous histological picture: note the presence of lymphocytic infiltrate, clusters of melanophages, lamellar fibroplasia, and neoangiogenesis without any invasive visible component of neoplastic cells (hematoxylin–eosin, original magnification 10×).

### 3.2. Prognostic Information

Regarding the reporting of the regression parameter in MM, as already said, the latest version of the recommendations of the CAP recommends reporting the parameter in the histopathological report of the MM, using ≤75% or >75% regression as a cut-off. In that sense, this scheme has largely replaced the previous ways of reporting regression and would appear to reduce inter-observer variability [4,13]. Regarding the staging of the regression, a paper by Kang et al. [37] proposed a three-step scheme divided into early, intermediate, and late stages of regression. Several melanophages are present in a dense inflammatory infiltrate in the early stages. Although inflammation is less pronounced than in the early stage, there is fibrosis with neoplastic nests in the intermediate stage. All neoplastic cells are replaced by fibrosis or tightly packed melanophages in the late stage. Although the inter-rater reliability of the three-stage classification approach may have declined, many dermatopathologists continue to utilize it [34]. The three-stage classification technique may be more suitable for MIS because it does not evaluate regression primarily based on a dermal component.

### 3.3. Treatment

A recent paper by Cartron et al. satisfactorily addressed the paradox of the definition "melanoma in situ with regression" [34]. In fact, in that work, the authors define how in the literature there are no systematic studies with large case series that address whether and how the presence of any regression, in the MIS setting, can influence the prognosis of patients. In our review of the literature, there are no studies that analyze this question in detail. Therefore, it is vital to address this topic in detail in order to understand whether MIS cases with regression phenomena should be handled differently from pure MIS cases.

### 3.4. Conclusive Considerations

As stated above, it is important to consider how misleading the diagnosis of MIS with regression can be. In fact, various studies have shown that regression, regardless of whether it is prognostically favourable, neutral, or unfavourable, represents a response by the host's immune system to the invasion attempt by MM cells. Therefore, it is reasonable to postulate that in carcinogenesis and invasion, MM is attacked by T lymphocytes, entailing the possibility of partial and/or total obliteration, the presence of clustering melanophages, and phenomena of sub-epithelial neoangiogenesis. All this would reveal that neoplastic cells were once present in that area of skin, which the regression has, in some way, attacked and resolved. Furthermore, regression measurement in mm should not be reported in addition to the Breslow thickness value, as it has the potential to upstage the neoplasm.

In consideration of the absence of studies with large case series, of reviews with meta-analyses, and, therefore, of a broad scientific consensus, expressions including "melanoma in situ with regression" should be avoided in the histopathological report and should be replaced with clearer and more exhaustive definitions, such as "melanoma with extensive regression phenomena and residual melanoma in situ component."

**Author Contributions:** Conceptualization, G.C. and A.C.; methodology, G.C.; software, G.I.; validation, G.C.; formal analysis, A.C.; investigation, G.C.; resources, A.C.; data curation, G.I.; writing—original draft preparation, G.C.; writing—review and editing, G.I.; visualization, A.C.; supervision, G.C. All authors have read and agreed to the published version of the manuscript.

**Funding:** This research received no external funding.

**Institutional Review Board Statement:** Not applicable.

**Informed Consent Statement:** Not applicable.

**Data Availability Statement:** Not applicable.

**Acknowledgments:** In memory of Antonietta Cimmino (A.C.).

**Conflicts of Interest:** The authors declare no conflict of interest.

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
