# Peer review of "Is It Time to Supersede the Diagnostic Term “Melanoma In Situ with Regression?” A Narrative Review"

_dermatopathology, doi:10.3390/dermatopathology10010018_

Round 1

Reviewer 1 Report

The authors conducted a literature review for studies of melanoma in situ (MIS) and regression and found no systematic studies with large case series to address the prognostic value of regression in the setting of MIS. Therefore the authors suggested that “MIS with regression” should be avoided in histopathologic reports.

The review needs to expand the introduction and discussion of controversial findings in literature. Authors should expand the result section by summarizing more details of studies in literature, giving information such as criteria used in different studies, definitions of regression, staging/grading methods, results and conclusions. The results may be better presented in a table.

The term of “MIS with regression” may include two situations: 1. pure MIS undergoing partial or complete regression. 2. MIS overlying dermal changes of regression (fibrosis, inflammation, melanophages, etc) which may represent a previous invasive component.

The impact of regression on prognosis indeed has been controversial in literature. Metastasis has been reported in patients with MIS and extensive regression. Studies specifically investigating the prognostic effect of regression on MIS are needed. Instead of avoiding reporting regression in cases of MIS as suggested by the authors, standardized reporting of regression in cases of MIS is important for future studies and to guide evidence-based practice.

Author Response

The authors conducted a literature review for studies of melanoma in situ (MIS) and regression and found no systematic studies with large case series to address the prognostic value of regression in the setting of MIS. Therefore the authors suggested that “MIS with regression” should be avoided in histopathologic reports.

The review needs to expand the introduction and discussion of controversial findings in literature. Authors should expand the result section by summarizing more details of studies in literature, giving information such as criteria used in different studies, definitions of regression, staging/grading methods, results and conclusions. The results may be better presented in a table.

Answer n’1: Dear Reviewer n’1, first of all, thank you very much for your useful tips to improve the quality of our manuscript. We have done all and we hope that, now, the paper will be fine.

The term of “MIS with regression” may include two situations: 1. pure MIS undergoing partial or complete regression. 2. MIS overlying dermal changes of regression (fibrosis, inflammation, melanophages, etc) which may represent a previous invasive component.

The impact of regression on prognosis indeed has been controversial in literature. Metastasis has been reported in patients with MIS and extensive regression. Studies specifically investigating the prognostic effect of regression on MIS are needed. Instead of avoiding reporting regression in cases of MIS as suggested by the authors, standardized reporting of regression in cases of MIS is important for future studies and to guide evidence-based practice.

Answer n’2: Thank you very much for these important tips and thoughts. We have decided not only to implement our reflections with these thoughts of yours but also to propose and adjust our conclusions according to this view. We thank you again.

Reviewer 2 Report

The paper should be a narrative review on the topic of regression in melanoma in situ and how this phenomenon should be considered in a therapeutic and prognostic perspective. However, little attention has been played on the phenomenon of regression and on the limitation in assessing it.  Moreover, 

Results, lines 67-68:   would you better specify inclusion criteria? 

Results, lines 82-89.  Histopathologic  criteria of MIS are well established. thus these lines may be omitted.

Results, lines 93-97: the sentence is obscure, especially regarding the concept of dysplastic melanocytes which appear innocent... 

Results, lines 97-101. Hallmarks of nested melanoma in situ are large junctional nests of different sizes, with bridging and cytonuclear atypias together with lesion asymmetry. 

Conclusive considerations, lines 134-135: the authors should propose better definitions for MIS with regression. 

Author Response

The paper should be a narrative review on the topic of regression in melanoma in situ and how this phenomenon should be considered in a therapeutic and prognostic perspective. However, little attention has been played on the phenomenon of regression and on the limitation in assessing it.  Moreover, 

Answer n’1: Dear Reviewer n’2, thank you very much for your kind tips useful to improve the quality of our manuscript. We have completed the paper in its entirety, focusing on aspects of regression, its definition, grading/staging methods, with particular attention to the difficulties of inter-observer homogenisation and with sound proposals for the future.

Results, lines 67-68:   would you better specify inclusion criteria? 

Answer n’2: Done, thank you very much.

Results, lines 82-89.  Histopathologic  criteria of MIS are well established. thus these lines may be omitted.

Answer n’3: Ok, we only left a few hints of definition.

Results, lines 93-97: the sentence is obscure, especially regarding the concept of dysplastic melanocytes which appear innocent... 

Answer n’4: Ok, Indeed, it might have been a somewhat obscure sentence: therefore, in the process of changing and revising the paper, we have changed this part.

Results, lines 97-101. Hallmarks of nested melanoma in situ are large junctional nests of different sizes, with bridging and cytonuclear atypias together with lesion asymmetry. 

Answer n’5: Yes, we better specificy.

Conclusive considerations, lines 134-135: the authors should propose better definitions for MIS with regression. 

Abswer n’6: Done, thank you.

Reviewer 3 Report

Congratulations on your systematic review of in-situ melanoma regression. Your article raises some interesting points, and the conclusion seems appropriate. However, I have some comments on the manuscript.

From a pathological perspective, I would like to see more discussion on the peritumoral microenvironment, such as papillary dermis fibrosis, inflammatory cell infiltration, and keratinocyte changes around regressed in-situ melanoma. It would be interesting to know whether the tumor cells regress spontaneously, due to inflammatory cell attack, or other signaling pathways.

Author Response

Congratulations on your systematic review of in-situ melanoma regression. Your article raises some interesting points, and the conclusion seems appropriate. However, I have some comments on the manuscript.

From a pathological perspective, I would like to see more discussion on the peritumoral microenvironment, such as papillary dermis fibrosis, inflammatory cell infiltration, and keratinocyte changes around regressed in-situ melanoma. It would be interesting to know whether the tumor cells regress spontaneously, due to inflammatory cell attack, or other signaling pathways.

Answer n’1: Dear Reviewer n’3, thank you very much for your helpful advice to improve the quality of our manuscript. We have therefore implemented the entire structure of the paper with all these arguments and, in particular, with updated references to possible pathways mediating melanoma regression. Thank you again.

Round 2

Reviewer 1 Report

No convincing studies to data suggest that the presence of regression in MIS necessitates wider margins. Evidence based practice relies on studies investigating the prognostic effect of regression on MIS.  Therefore recognizing the presence of regression in cases of MIS at clinical and histopathological ground retains value.

In results section, page 4, line 125, the references of these 8 articles should be listed here.

Majority of this review summarized histologic features of MIS and regression as shown in section 3.1, followed by 3.2 conclusive consideration. It is better to category and create 3.2 for prognostic information 3.3 for treatment and 3.4 for conclusion. Contents and key points of these 8 articles should be discussed in these sections.

Page 4, line 153, “dysplastic melanocytes” better to change to “atypical melanocytes”.” Dysplastic” is usually used for “dysplastic nevus” by dermatopathologists.

Page 4, line 157, “typical/typical” do you mean “typical/atypical”?

Page 4, line 164, (Figure 1-3) do you mean (Figure 2-4)?

Page 5, line 174 “stading” do you mean “staging”

Figure 2 and Figure 3 are taken from the same case. Figure 3 is not “Another example”. Figure 2-4 show an invasive melanoma case with stromal changes of regression. It’s better to show a melanoma in situ case with stromal changes of regression.

Author Response

Reviewer n’1: No convincing studies to data suggest that the presence of regression in MIS necessitates wider margins. Evidence based practice relies on studies investigating the prognostic effect of regression on MIS.  Therefore recognizing the presence of regression in cases of MIS at clinical and histopathological ground retains value.

Answer n’1: dear Reviewer N’1,

thank you for your comments aimed at improving the quality of our manuscript. We understand your point of view and believe it is absolutely shareable; we would therefore like to clarify that in our view, based on the data coming out of this narrative review, it is not only important to maintain and report regression in cases of MIS, but no longer to report it as such in the report, but rather (as proposed at the end of the paper) to use the term "melanoma with extensive regression and residual in situ component", symbolising the possibility that regression is already a later stage than MIS.

Reviewer n’1: In results section, page 4, line 125, the references of these 8 articles should be listed here.

Answer n’2: Done, thank you very much.

Reviewer n’1: Majority of this review summarized histologic features of MIS and regression as shown in section 3.1, followed by 3.2 conclusive consideration. It is better to category and create 3.2 for prognostic information 3.3 for treatment and 3.4 for conclusion. Contents and key points of these 8 articles should be discussed in these sections.

Answer n’3: Done, thank you very much.

Reviewer n’1: Page 4, line 153, “dysplastic melanocytes” better to change to “atypical melanocytes”.” Dysplastic” is usually used for “dysplastic nevus” by dermatopathologists.

Answer n’4: Sure, thank you very much.

Reviewer n’1: Page 4, line 157, “typical/typical” do you mean “typical/atypical”?

Answer n’5: yes, sorry. We corrected it.

Reviewer n’1: Page 4, line 164, (Figure 1-3) do you mean (Figure 2-4)?

Page 5, line 174 “stading” do you mean “staging”

Figure 2 and Figure 3 are taken from the same case. Figure 3 is not “Another example”. Figure 2-4 show an invasive melanoma case with stromal changes of regression. It’s better to show a melanoma in situ case with stromal changes of regression.

Answer n’6: thank you very much for these useful tips. We have changed all according to your advices. Thanks again.

Reviewer 3 Report

Thank you for your response to the previous comments. However, there are several issues that need to be addressed in the manuscript.

Firstly, there is inconsistency between the abstract (13 articles included) and figure 1/result section (only 8 papers included).

Additionally, there are several spelling errors and inconsistent statements that need to be corrected, such as "lymphocitic" to "lymphocytic," "dermic" to "dermis," and "nucleu" to "nuclei." I recommend using appropriate software functions to detect and correct all misspellings and grammar errors.

Author Response

Reviewer n'2:

Thank you for your response to the previous comments. However, there are several issues that need to be addressed in the manuscript.

Firstly, there is inconsistency between the abstract (13 articles included) and figure 1/result section (only 8 papers included).

Additionally, there are several spelling errors and inconsistent statements that need to be corrected, such as "lymphocitic" to "lymphocytic," "dermic" to "dermis," and "nucleu" to "nuclei." I recommend using appropriate software functions to detect and correct all misspellings and grammar errors.

Answer n'1: Dear Reviewer n'2, thank you very much for these useful tips to improve quality of our manuscript. We changed according to your suggestions.